# Observation of geometric phase effect through backward angular oscillations in the H + HD → H₂ + D reaction

**Shihao Li[1,7], Jiayu Huang [2,3,7], Zhibing Lu[1], Yiyang Shu[1], Wentao Chen[1], Daofu Yuan[1], Tao Wang [4], Bina Fu [2,5], Zhaojun Zhang [2,5,6] ✉, Xingan Wang [1,5] ✉, Dong H. Zhang [2,4,5] ✉ & Xueming Yang [2,4,5] ✉**

Quantum interference between reaction pathways around a conical intersection (CI) is an ultrasensitive probe of detailed chemical reaction dynamics. Yet, for the hydrogen exchange reaction, the difference between contributions of the two reaction pathways increases substantially as the energy decreases, making the experimental observation of interference features at low energy exceedingly challenging. We report in this paper a combined experimental and theoretical study on the H + HD → H₂ + D reaction at the collision energy of 1.72 eV. Although the roaming insertion pathway constitutes only a small fraction (0.088%) of the overall contribution, angular oscillatory patterns arising from the interference of reaction pathways were clearly observed in the backward scattering direction, providing direct evidence of the geometric phase effect at an energy of 0.81 eV below the CI. Furthermore, theoretical analysis reveals that the backward interference patterns are mainly contributed by two distinct groups of partial waves ($J \sim 10$ and $J \sim 19$). The well-separated partial waves and the geometric phase collectively influence the quantum reaction dynamics.

Geometric phase (GP) is an important concept that originates from the presence of a conical intersection (CI) in a molecular system. When the said molecular system encircles the CI, its adiabatic electronic wavefunction undergoes a sign change. Since the total wave function is single-valued, a compensating sign change must be introduced in the nuclear wavefunction, which gives rise to the GP. The GP effect was originally discovered independently by Pancharatnam and Longuet-Higgins in the 1950s[1,2], and was further generalized to all adiabatic processes by Berry in 1984[3]. Over the past several decades or so, the influence of GP has become a vital topic in various fields in condensed matter physics and chemical physics[4–7].

Tremendous efforts have been made in the studies on the involvement of GP in elementary chemical reactions[8–10], with a special focus on the hydrogen exchange reaction. In this reaction, the H₃ triatomic molecular system has a well-known CI connecting its electronic ground state and the first excited state potential energy surfaces at the equilateral triangle (D₃ₕ) geometry[11–15], which makes the H + H₂ reaction and its isotopic variants ideal prototypes for studying the GP effect. As first shown by Mead and Truhlar, the effect of the GP could be included in adiabatic quantum dynamical calculations by introducing an effective vector potential in the nuclear motion Hamiltonian[16–19]. They proposed that the GP changes the sign of the interference term between the reactive and nonreactive scattering in the H + H₂ reaction, thereby

[1]Department of Chemical Physics, University of Science and Technology of China, Hefei 230026, China. [2]State Key Laboratory of Molecular Reaction Dynamics, Dalian Institute of Chemical Physics, Chinese Academy of Sciences, Dalian 116023, China. [3]Department of Physics, Dalian University of Technology, Dalian 116024, China. [4]Department of Chemistry, College of Science, Southern University of Science and Technology, Shenzhen 518055, China. [5]Hefei National Laboratory, Hefei 230088, China. [6]University of Chinese Academy of Sciences, Beijing 100049, China. [7]These authors contributed equally: Shihao Li, Jiayu Huang. ✉e-mail: zhangzhj@dicp.ac.cn; xawang@ustc.edu.cn; zhangdh@dicp.ac.cn; xmyang@dicp.ac.cn

exerting a substantial impact on the differential cross sections (DCSs) and integral cross sections (ICSs). In the 2000s, Kendrick performed a series of quantum reactive scattering calculations for the $H + H_2$ reaction and its isotopic variants at a wide range of collision energies[20–23]. The calculations revealed rigorous cancellation of GP effects when product distributions are summed over all partial waves to give the DCSs and ICSs. On the other hand, Althorpe and coworkers carried out quantum wave-packet calculations and discovered that the GP effect could result in a slight shift of the rapid oscillations which are superposed on the main envelope in sideways scattered products in $H + H_2$ ($v = 1, j = 0$) reaction at a total energy of 2.30 eV[24]. The significance of the GP effect at various different energies has also been studied[25–28].

In an effort to experimentally observe the GP effect in the hydrogen exchange reaction, various well-designed experiments have been carried out using the photoinitiated reaction analyzed by the law of cosines (PHOTOLOC) method and the H-atom Rydberg tagging time-of-flight (HRTOF) technique[29–33]. However, there was no clear evidence showing that the GP could influence the reaction until the last five years. Since 2018, high-resolution crossed molecular beams experiments were carried out by using the time-sliced velocity map ion imaging technique and the near-threshold ionization method. Yuan et al. studied the $H + HD \rightarrow H_2 + D$ reaction at the collision energies of 2.77 and 2.28 eV (about 0.25 eV above and below the CI, respectively). The observed high-resolution DCSs with fast angular oscillations in the

forward scattering direction unambiguously reveal the influence of the GP effect near the CI[34,35]. Meanwhile, Xie et al. employed the D-atom Rydberg tagging TOF method combined with the time-dependent quantum wave-packet approach to study $H + HD \rightarrow H_2 + D$ reaction[36]. Oscillatory structure was observed in backward scattered signals (180 degrees in the center of the mass frame) as a function of collision energies between 1.94 and 2.21 eV. These results clearly show that quantum interference between reactive pathways around a CI serves as an ultrasensitive probe on chemical reaction dynamics.

The observation and identification of quantum interference features have played a crucial role in understanding the GP effect in hydrogen exchange reactions. The interference features strongly depend on collision energy and scattering angle, and can be significantly enhanced when the contributions of the two pathways are nearly identical. However, the accurate calculation predicted that the roaming insertion mechanism pathway has about two orders of magnitude smaller reactivity than the direct abstraction reactive pathway when the collision energy is lower than 1.80 eV. This big contrast makes the experimental investigation on the interference extremely challenging thus hindering the understanding of detailed quantum reaction dynamics. As a result, particular efforts are required for experimentally probing the GP effect at a low collision energy. Here we report a combined experimental and theoretical study on the $H + HD$ reaction at the collision energy of 1.72 eV (0.81 eV below CI). By using the velocity map imaging technique with the near-threshold ionization technique utilized in the crossed molecular beams experiment, quantum state resolved DCSs with high resolution in both angular and energy distribution were obtained. In particular, angular oscillations in the backward scattering direction with a period of ~10° have been observed in rotational state-specific DCS. Together with the accurate theoretical calculations, we unambiguously discovered these backward fine angular structures are a new manifestation of the GP effect in the hydrogen exchange reaction. Partial waves responsible for backward scattering oscillations are also thoroughly analyzed, providing essential information on the quantum reaction dynamics surrounding the CI.

## Results and discussion
### High-resolution experimental image
As shown in Fig. 1a, a high-resolution product D-atom image for the $H + HD \rightarrow H_2 + D$ reaction at 1.72 eV was obtained in the crossed molecular beam experiment. The quantum states of co-product $H_2$ are clearly resolved in DCSs for a series of ro-vibrational states $H_2$ ($v', j'$). Based on the measured image, quantum state-resolved DCSs were derived (the data analysis method is shown in Supplementary Note 2) and shown in Supplementary Fig. 2. The theoretically simulated product D-atom image is shown in Fig. 1b, which was calculated by using the adiabatic quantum dynamics method with GP included. The experimental and theoretically simulated images show very good agreement.

### The backward angular oscillatory patterns and GP identification
Figure 2 shows the backward angular distributions of co-product $H_2$ with quantum states, corresponding to product $H_2$ ($v' = 0$, $j' = 9$ and $v' = 1$, $j' = 3$), $H_2$ ($v' = 1$, $j' = 9$ and $v' = 2$, $j' = 3$), and $H_2$ ($v' = 0$, $j' = 11$ and $v' = 1$, $j' = 7$), respectively, marked by black arrows in Fig. 1. It is worth noting that the energy differences between the two states shown in each panel of Fig. 2 are small, and the signals are merged into adjacent pixels in the image. We therefore used the "&" connecting these two ro-vibrational states with similar energies (please see data analysis in the Supplementary Note 1 and Supplementary Fig. 1). Figure 2a shows the angular distribution for ($v' = 0$, $j' = 9$) and ($v' = 1$, $j' = 3$) states. The experimental shoulder structure at around 175° strongly deviates from the theoretical NGP results but is in very good agreement with the theoretical GP result. Fast oscillations were observed as well for $H_2$

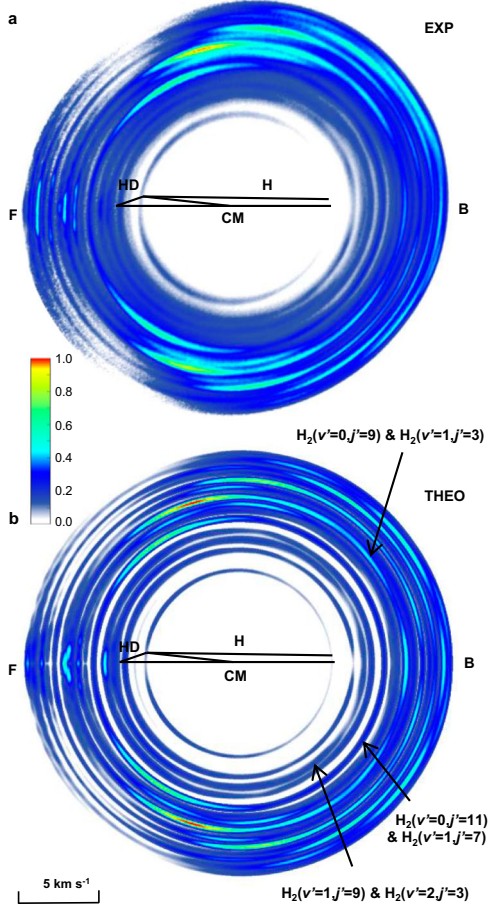

**Fig. 1 | The product D-atom images. a** Experimental (EXP) and **b** theoretically (THEO) simulated images of product D from the $H + HD$ ($v = 0, j = 0$) $\rightarrow H_2$ ($v'$, $j'$) + D reaction at Ec = 1.72 eV. "F" and "B" represent the forward and the backward scattering direction, respectively. The symbols "HD" and "H" represent the velocity vectors of the HD molecular beam and the H-atom beam, respectively. "CM" represents the origin of the center-of-mass coordinate system. Source data are provided as a Source Data file.

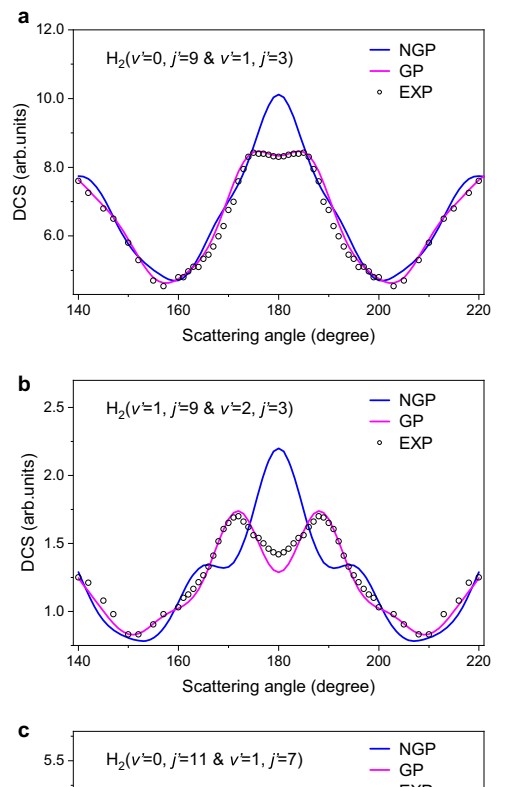

**Fig. 2 | The backward angular distribution. a** for $H_2$ ($v' = 0, j' = 9$ and $v' = 1, j' = 3$), **b** for $H_2$ ($v' = 1, j' = 9$ and $v' = 2, j' = 3$) and **c** for $H_2$ ($v' = 0, j' = 11$ and $v' = 1, j' = 7$). The experimental data is displayed with hollow circles (EXP). The theoretical results depicted by the blue lines (NGP) do not incorporate the geometric phase, whereas the theoretical results represented by the magenta lines (GP) do incorporate the geometric phase. Source data are provided as a Source Data file.

($v' = 0, j' = 3, 5$) states in the forward scattering direction, but the forward structures remain unaffected by the GP at the studied collision energy (shown in Supplementary Fig. 2). As a result, it is clear that for these states, the influence of GP at 1.72 eV is mainly manifested near the backward scattering angular range. Figure 2b shows the angular distribution for the ($v' = 2, j' = 3$) and ($v' = 1, j' = 9$) states. Interestingly, there is a distinguished peak structure at 172° in the backward scattering direction. Theoretical calculations reveal that the incorporation of the GP manifests a double peak feature in the backward direction. Furthermore, the findings presented in Fig. 2c exhibit an even more complicated picture. Two oscillatory peaks of DCS, with an oscillation period of about 11°, are observed in the angular distribution for the ($v' = 0, j' = 11$) and ($v' = 1, j' = 7$) states. The first peak is near 153°, and the second peak is at 164°. A systematic deviation relative to NGP results in the angular distribution for the $H_2$ ($v' = 0, j' = 11$ and $v' = 1, j' = 7$) state, as shown clearly in Fig. 2c. At this specific low collision energy, the influence of GP has been clearly identified through the oscillations in the backward scattering angular range.

To explore the underlying dynamics of observed oscillations, we performed quasi-classical trajectory calculations (see Supplementary

Note 4). We ran a total of 50 million trajectories at a collision energy of 1.72 eV, resulting in a statistical error of only 0.05% in the total reactive cross section. Out of these trajectories, 3.597 million exhibited reactivity, leading to the production of $H_2$ products. The majority, 99.912%, followed path 1, while only a small fraction, 0.088% or 3165 trajectories, chose path 2. The first pathway follows the well-established direct abstraction mechanism, while the second is an unusual roaming insertion pathway. The observed peak structure in backward scattering should be attributed to the quantum interference occurring between path 1 and 2 surrounds the CI. The unusual roaming insertion pathway was studied in a prior work[36] by scanning the collision energy at the scattering angle of 180 degrees. However, its high-resolution angular distribution has not been studied and the influence of GP on state-specific backward DCS remains unknown.

## Reaction mechanisms investigation

In order to illustrate the backward angular distributions of different reaction pathways, we extracted the scattering amplitudes from path 1 and path 2 through a straightforward topological approach proposed by Althorpe and coworks[24]. The wave function is decomposed through a topological approach where $\Psi^\pm = 1/\sqrt{2}(\Psi^{path1} \pm \Psi^{path2})$, and $\Psi^+ = \Psi^{NGP}$ and $\Psi^- = \Psi^{GP}$. The scattering amplitudes from path 1 and path 2 are given by the general expression:

$$f_{path1} = \frac{1}{\sqrt{2}}\left(f_{NGP} + f_{GP}\right) \qquad (1)$$

$$f_{path2} = \frac{1}{\sqrt{2}}\left(f_{NGP} - f_{GP}\right) \qquad (2)$$

The square moduli of the scattering amplitudes give the DCSs $\sigma^\pm$ as

$$\sigma^\pm = \frac{1}{2}\left(\left|f_{path1}\right|^2 + \left|f_{path2}\right|^2 \pm 2\left|f_{path1}\right|\left|f_{path2}\right|\cos(\Delta)\right) \qquad (3)$$

where $\Delta$ represents the phase difference between path 1 and path 2. The inclusion of GP alters the phase difference and thus changes the scattering amplitudes of DCSs, which manifested as the GP effects in the present experimental image. Figure 3 presents the full angular range theoretical DCSs for $H_2$ products ($v' = 0, j' = 9$), ($v' = 1, j' = 3$), ($v' = 1, j' = 9$), ($v' = 2, j' = 3$), ($v' = 0, j' = 11$) and ($v' = 1, j' = 7$) from path 1 and path 2 at 1.72 eV. It is clear that scattering amplitudes of path 2 is nearly two orders of magnitude smaller than path 1, showing the difficulties in probing the GP effects experimentally. It is obvious that the angular distributions for the two reaction pathways are very different. Path 1 leads to an angular distribution relatively small in backward direction, whereas path 2 leads to slightly dominant backward scattering. More interestingly, the oscillation period in path 2 changes noticeably faster than in path 1, indicating the presence of distinct key partial waves influencing the two reaction pathways.

To better understand the physical origin of these fine oscillatory structures, we performed a detailed dynamics analysis. The ro-vibrational state-resolved DCS for the reaction with the initial state ($v_0, j_0, K_0$) is given by:

$$\frac{d\sigma_{v'j' \leftarrow v_0 j_0}(\theta, E)}{d\Omega} = \frac{1}{(2j_0 + 1)}\sum_{K_0}\sum_{K'}\left|\frac{1}{2ik_{v_0 j_0}}\sum_J (2J+1) \times d^J_{K'K_0}(\theta)S^J_{v'j'K' \leftarrow v_0 j_0 K_0}(E)\right|^2$$
$$= \frac{1}{(2j_0 + 1)}\sum_{K_0}\frac{d\sigma_{v'j' \leftarrow v_0 j_0 K_0}(\theta, E)}{d\Omega}$$

$$(4)$$

where $J$ represents the system's total angular momentum, $K'$ ($0 \le K' \le$ min ($j', J$)) denotes the projection of $J$ (or $j'$) onto the recoil direction,

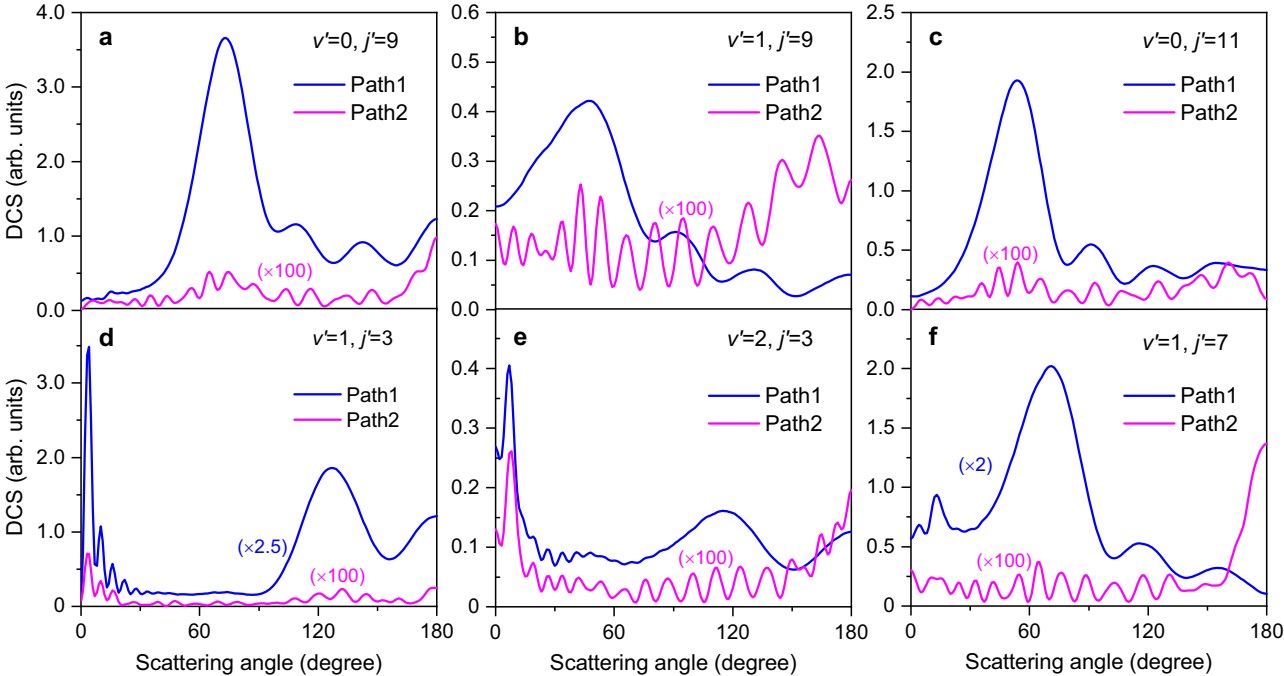

**Fig. 3 | Relative differential cross sections from Path 1 and Path 2. a** for product $H_2$ ($v'=0, j'=9$), **b** ($v'=1, j'=9$), **c** ($v'=0, j'=11$), **d**, ($v'=1, j'=3$), **e** ($v'=2, j'=3$), and **f** ($v'=1, j'=7$). "Path 1" and "Path 2" represent two distinct reactive pathways: the direct abstraction reaction (Path 1) and the unusual roaming insertion pathway (Path 2), respectively. Source data are provided as a Source Data file.

$d^J_{K'K}(\theta)$ represents the Wigner d-matrix. By using this approach, we calculated the $K'$ quantum-number-specific DCSs at a collision energy of 1.72 eV for the reaction occurring along path 1 and path 2. From the data presented in Fig. 3, it is evident that the combination DCSs are predominantly influenced by $H_2$ products level ($v'=0, j'=9$), ($v'=2, j'=3$) and ($v'=0, j'=11$) in the backward direction.

Among these three product states, the backward DCSs for the $H_2$ ($v'=2, j'=3$) product almost entirely comes from the $K'=0$ components (shown in Supplementary Fig. 3), thereby providing a solid proof for experimental observation of the influence of GP on the $K'$ quantum-number-specific DCSs. The helicity-specific angular distribution for the $H_2$ products ($v'=0, j'=9$) and ($v'=0, j'=11$) are illustrated in Supplementary Fig. 4 and Fig. 4, respectively.

It is quite clear that the backward angular oscillatory structure for products ($v'=0, j'=9$) and ($v'=0, j'=11$) via path 1 are predominantly contributed by partial wave $J=9$ and 10, while products via path 2 are mainly contributed by partial wave $J=15$, 16, and 19. Based on classical collision theory, the scattering amplitudes can be roughly split into low-impact ($0 < J \leq 10$) and high-impact ($10 < J \leq 40$) parameter contributions, representing head-on and glancing collisions, repectively[37]. Therefore, products from path 1 and path 2 are intrinsically related to two distinct groups of partial waves that correspond to different collision mechanisms. As shown in Fig. 5, there are two schematic trajectories with low and high impact parameters through two paths in the H + HD ($v=0, j=0$) → $H_2$ ($v', j'$) + D reaction (Fig. 5a and c) that correspond to different reactive trajectories for two paths surrounding the CI (displayed in hyperspherical coordinates in Fig. 5b and d). Path 1 is the well-known direct abstraction mechanism, and the mechanism of path 2 is much subtler. The incoming H' atom approach HD molecular and passes the first H'-D-H linear transition state. As the HD molecule stretches, the H' atom inserts between the H-D bond by experiencing the second D-H'-H transition state. It is worth noting that products from path 2 are mainly formed via the glancing collision process and scatter to the backward direction, providing direct evidence for the roaming-insertion mechanism in the quantum mechanical picture. The well-separated two distinct groups of partial waves that move along the direct abstraction pathway and the insertion roaming pathway and the geometric phase collectively influence the quantum reaction dynamics.

We have successfully observed the backward angular oscillatory patterns resulting from the interference between direct abstraction mechanism pathway and roaming-insertion reactive pathway. Notably, this is a distinct manifestation of the GP effect in the DCSs and was discovered at an energy of 0.81 eV below the CI. The findings demonstrate that the quantum state-specific fine angular oscillations act as a highly responsive indicator of the GP effect. Furthermore, the theoretical analysis reveals that the backward interference patterns are mainly contributed by two distinct groups of partial waves ($J \sim 10$ and $J \sim 19$), which move along the direct abstraction pathway and the insertion roaming pathway, respectively. The well-separated partial waves and the geometric phase that evolves through the two reaction pathways collectively influence the quantum reaction dynamics, revealing new insights into complex quantum dynamical mechanisms at low collision energy.

## Methods
### Experimental setup
In this study, the hydrogen atom beam was produced through the photodissociation of hydrogen bromide (HBr) molecules using the fifth harmonic (213 nm) of a Nd: YAG laser. The intensity of photolysis light is 25-30 mJ/pulse, and the HBr molecular beam was produced from a pulsed valve (Parker valve) that is 10.0 cm below the photolysis laser beam. By configuring the polarization of the photolysis laser (213 nm) in a vertical orientation, the faster hydrogen atom beam with the velocity of 19.80 km/s was specifically chosen. The HD molecules were held at the temperature of liquid nitrogen and were subsequently supersonically expanded by an Even-Lavie valve (Lamid). The HD molecule beam exhibits a velocity of 1.24 km/s, with approximately 97% of its components existing in the ground rovibrational state[38,39]. The collision energy was 1.72 eV with the crossing angle of 160° between the H atom beam and HD molecular beam. The product D

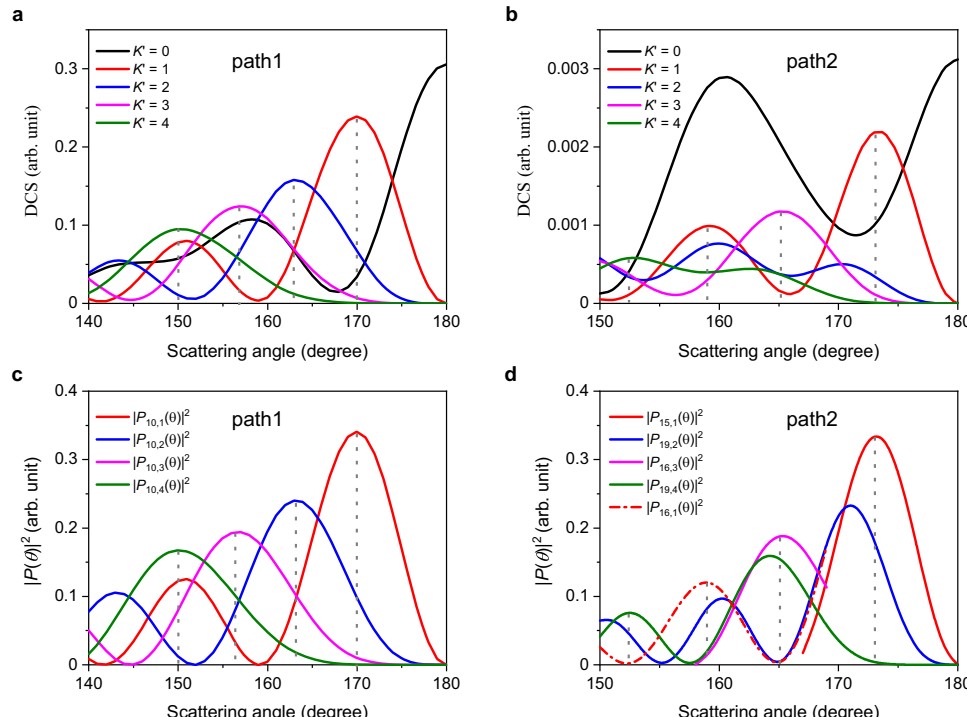

**Fig. 4 | Theoretical analysis for product H₂ (*v′* = 0, *j′* = 11) generated through Path 1 and Path 2 in the backward scattering direction. a** *K′* quantum-number-specific DCSs for Path 1 product, *K′* is the projection of *J* (or *j′*) onto the recoil direction (the helicity quantum number). **b** *K′* quantum-number-specific DCSs for Path 2 product. **c** Square moduli of the associated Legendre polynomials $|P_{J,K}(\theta)|^2$

with *J* = 10 and *K* = 1, 2, 3, 4. **d** Square moduli of $|P_{J,K}(\theta)|^2$ with *J* = 15, 16, 19 and *K* = 1, 2, 3, 4. "Path 1" and "Path 2" represent two distinct reactive pathways: the direct abstraction reaction (Path 1) and the unusual roaming insertion pathway (Path 2), respectively. The vertical dotted lines denote the peak positions of the corresponding curves. Source data are provided as a Source Data file.

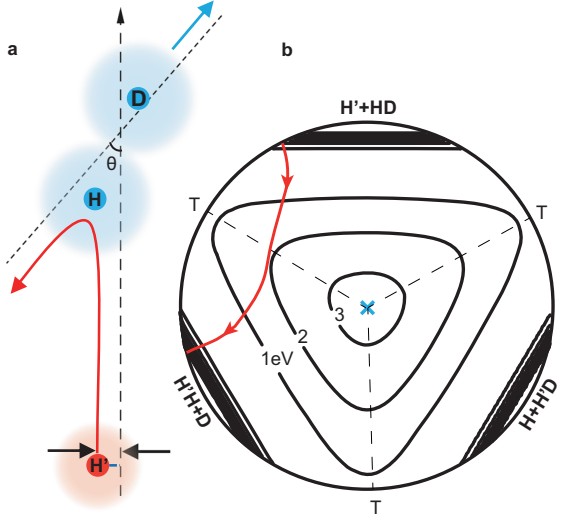

**Low-impact parameter through path 1**

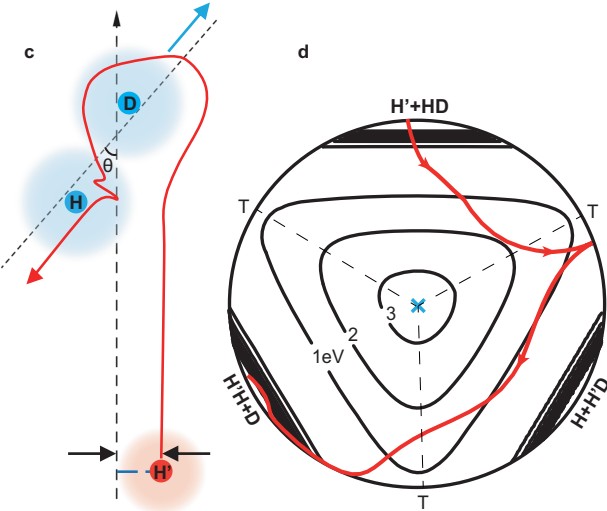

**High-impact parameter through path 2**

**Fig. 5 | Analysis on two distinct groups of partial waves that move along the direct abstraction pathway and the insertion roaming pathway. a, c** Present two schematic trajectories with differing impact parameters through two paths in the H + HD (*v* = 0, *j* = 0) → H₂ (*v′*, *j′*) + D reaction. Directions of incident H atom and outcome D atom are marked by upward black arrows and blue arrows, respectively.

The HD/H reactants are schematically represented by blue and red shades, respectively. **b, d** Depict the corresponding reactive classical trajectories for two paths in hyperspherical coordinates. with black dotted lines denoting transition states (T) and the blue crosses marking the locations of the CIs.

generated from the H + HD reactive scattering underwent ionization using the 1 + 1′ (121.6 nm + 364.5 nm) near threshold resonance-enhanced multiphoton ionization (REMPI) method. The ionized products were guided to the MCP detector by ion optics, and the images

were recorded by a CCD camera. A real time ion event counting method was applied during the data acquisition of ion images. In order to achieve the equal detection efficiency of D atom products with different velocities, the wavelength of the probe laser was scanned

back and forth to tune the wavelength of the VUV laser beam to cover the whole Doppler profile of the D atom products during the measurement.

## Theoretical method

We performed adiabatic quantum dynamics calculations on the BKMP2 PES[40] and extracted state-to-state dynamical information by using the product-coordinate-based wave packet method[41]. The GP is incorporated into wave packet calculations by employing the diabatic version of the vector potential approach[42]. The numerical parameters provided in the supplementary materials facilitate the convergence of DCSs for collision energies up to 2.5 eV.

## Data availability

Data supporting the findings of this study are available from the corresponding authors upon request. Additionally, source data are provided with this paper and can be accessed at https://doi.org/10.6084/m9.figshare.25020434. Source data are provided with this paper.

## Code availability

The code that supports the theoretical calculations in this paper is available from the corresponding authors upon request.

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

## Acknowledgements

This work was supported by the National Natural Science Foundation of China (Grant Nos. 22125302 (X.W.), 22322305 (Z.Z.), 22173097 (Z.Z.), 22103084 (J.H.), 22233003 (J.H.), and 22288201(D.Z.)), the Guangdong Science and Technology Program (Grant Nos. 2019ZT08L455 and 2019JC01X091) (X.Y.), the Innovation Program for Quantum Science and Technology (Grant No. 2021ZD0303304) (X.Y.), the Innovation Program for Quantum Science and Technology (Grant No. 2021ZD0303305) (D.Z.) and the Dalian Innovation Support Program (Grant No. 2021RD05) (D.Z.). We are gratefully indebted to Chang Luo and Yuxin Tan for their assistance in experiments and helpful discussions.

## Author contributions

X.Y., D.Z., X.W., and Z.Z conceived and supervised the research. The experiments were carried out by S.L., Z.L., Y.S., W.C., D.Y., and X.W. Data analysis and interpretation were performed by S.L., Z.L., Y.S., W.C., D.Y., T.W., X.W., and X.Y. Theoretical calculations were performed by J.H., Z.Z., B.F., and D.Z. The manuscript was written by X.Y., D.Z., X.W., and Z.Z., with contributions from all authors. All authors contributed to discussions about the content of the paper.

## Competing interests

The authors declare no competing interests.
