## [Peer Review File · Nature Communications]

REVIEWER COMMENTS

Reviewer #1 (Remarks to the Author):

“Observation of geometric phase effect through backward angular oscillations in the H + HD → H₂ + D reaction”

This combined experimental and theoretical work is simply outstanding, perhaps the very state-of-the-art in the field. It is amazing that still more pieces are being added to the puzzle of the H + H₂ exchange reaction. But this one, which may perhaps represents the last one, bringing to completion a masterpiece painting of the most fundamental of all chemical reactions, is a particularly important one, because deals in an important, rigorous manner with the “geometric phase (GP)” effect in this system. In particular, the paper addresses a fundamental issue: can geometric phase effects manifest themselves at energies “much lower” than the conical intersection between the ground and first excited H₃ PES?

In a previous excellent similar study (Yuan et al., *Science* 2018; ref. 34), the (essentially) same authors (Xingan Wang and Xueming Yang groups and their theoretical colleagues) reported a high-resolution crossed molecular beams study of the H + HD → H₂ + D reaction at a collision energy of 2.77 eV, which is somewhat above (by 0.23 eV) the conical intersection (CI). Velocity map ion imaging revealed fast angular oscillations in product quantum state-resolved differential cross sections (DCSs) in the forward scattering direction for H₂ products at specific rovibrational levels. The experimental results were found to agree with adiabatic quantum dynamical calculations only when the GP effect is included.

Later on (Yuan et al., *Nat. Comm.* 2020; ref. 35) a similar study at a collision energy of 2.28 eV on H + HD revealed GP effects considerably below the CI. At about the same time, Xueming Yang group (Xie et al., *Science* 2020; ref. 36) provided an interesting case of quantum interference between two topological pathways in the H+HD→H₂+D reaction in the collision energy range between 1.94 and 2.21 eV, manifested as oscillations in the energy dependent differential cross section for the H₂($v'=2$, $j'=3$) product in the backward scattering direction. The remarkable oscillation patterns observed were attributed to the strong quantum interference between the direct abstraction pathway and an unusual rebounding insertion pathway. Interestingly, the observed interference pattern also provided a sensitive probe of the geometric phase effect at energy far below the conical intersection in this benchmark system.

In the present manuscript, in what it is arguably the most detailed and highest resolution investigation of a reactive scattering process to date, the joint experimental/theoretical work on the H + HD reaction reveals new, exquisite details on the issue of conical intersection effects in the

most fundamental of all chemical reactions, at an energy (1.72 eV) much lower (by nearly 1 eV) than the CI. This really impressive (from both experimental and theoretical points of view) work witnesses the experimental and theoretical ingenuity of the authors and certainly well expresses the level of sensitivity and precision as well as of interpretation that can be achieved nowadays in collision experiments using state-of-the-art molecular beam/laser techniques and theoretical methodologies. After the two *Science* (2018 and 2020) and the *Nat. Comm.* (2020) papers quoted above, the present work reports a high-resolution crossed molecular beam study, similar to some extent to the work of 2018 and 2020, but with higher sensitivity (thanks to an improved version of the apparatus) and at a collision energy of only 1.72 eV, which is much lower than in the *Nat. Comm.* (2020) study and, specifically, 0.81 eV below the conical intersection. At this low energy the roaming insertion pathway is theoretically predicted to represent only a very small fraction (< 0.1%) of the overall contribution to the reactive signal. Yet, oscillatory structures arising from the interference of reaction pathways were very clearly observed in the backward scattering direction, providing, for the first time, direct evidence of the geometric phase effect at the energy of 0.81 eV below the conical intersection, something never imagined that could be experimentally observed before this study, and that could impact the quantum dynamics of the reaction.

The observed backward angular oscillatory patterns resulting from the interference between the direct abstraction mechanism pathways and the roaming-insertion pathways were very clearly observed and demonstrated to represent a manifestation of the GP effect in the differential cross section. An impressive, detailed theoretical analysis has revealed that the backward interference patterns are mainly contributed by two distinct groups of partial waves, which move along the direct abstraction pathways and the insertion roaming pathway. All this collectively influences the quantum reaction dynamics and reveals new insights into complex quantum dynamical mechanisms at low collision energy.

The manuscript is extremely well written. The introduction is truly excellent, the same the experiment, as well as the theory which is very comprehensive. All state-of-the-art. It should be noted that the experimentally reported and theoretically fully rationalized quantum interference patterns in the form of oscillations in the state-to-state differential cross section between two topological different reaction pathways in a chemical reaction has never been observed before at this low collision energy and neither theoretically predicted previously. The finding that this is a probe of the geometric phase effect far below the energy of the conical intersection in the H + H₂ reaction should be “framed and put in textbooks”. We can finally say that after nearly 90 years since

London, physical chemists have reached an even deeper (perhaps full?) understanding of the “simplest” chemical reaction.

This work is outstanding and certainly deserves publication in Nature Communications.

I have spotted only a few English imperfections that I am listing below, together with a few minor comments that the authors may wish to take in account.

1) page 3, line 48: “... concept that originated from ...” should be “... concept that originates from ...”. Line 60: “...prototypes in studying the GP effect.” Should be “...prototypes for studying the GP effect.”. Line 74: “... GP effect in hydrogen exchange reaction,...” should be “... GP effect in the hydrogen exchange reaction,...”.

2) page 4, line 96: “hinders the understanding of detailed quantum dynamics.” should be “hindering the understanding of the detailed quantum dynamics.”. Line 97: “required in experimental probing the GP effect” should be “required for experimentally probing the GP effect”. Line 98: “at a collision energy of 1.72 eV” should be “at the collision energy of 1.72 eV”. Line 101: “DCSs ... was obtained” should be “DCSs ... were obtained”.

3) Page 5, line 121: “the data analysis method shown in ...” should be “the data analysis method is shown in ...”. Line 123-124: “using the adiabatic quantum dynamics method with GP included.” I think that here the PES which has been used should be mentioned and referenced (BKMP2 PES).

4) Page 11, line 231: “approaches HD molecular” should be “approach the HD molecule”. Line 232: “HD molecular stretches” should be “HD molecule stretches”. Line 248: “we have...” should be “We have ...”. Line 249: “... between directed abstraction ...” should be “... between direct abstraction ...”.

5) Page 12, line 264: “of an Nd: YAG laser” should be “of a Nd:YAG laser”. Line 277: the BKMP2 PES is mentioned, but there is not a reference for it.

6) Page 13, line 326: BaÑares should be Bañares.

7) Supplementary information. Page 3, 5 lines from bottom: A rotational state-resolved TKER spectrum in the backward scattering direction is obtained.” I suggest to add at the end of the sentence (see Fig. 1). In fact, Fig. 1 is never mentioned in the text.

8) Supplementary information. Page 4, 1st line: “average valve” should be “average value”.

9) Supplementary information. Page 5: Figs. 3 and 4 are not introduced in the text.

Reviewer #2 (Remarks to the Author):

Reviewer #3 (Remarks to the Author):

After decades of effort, the geometric phase effect on elementary chemical reactions has only recently been documented, largely through work of some of these authors. However, such effects have been seen only at high energy in the region around the conical intersection (CI). In this beautifully-written manuscript, the present authors combine extraordinary experimental data obtained far below the CI, where two interfering paths have vastly different weights, with quantum scattering calculations to demonstrate clear evidence of the GP effect. The effect is seen in a narrow backscattered region for particular product states, and the contributing partial waves identified. It was a great pleasure to read this paper, and I certainly believe it merits publication in Nature Comm. I recommend publication as-is, although I have one small comment: I don't know what the reference to "And/@" means (l. 138) and I don't see a discussion of this in the SI. Perhaps they could clarify this.

Responses to the reviewers' comments

We would like to express our sincere thanks to all the reviewers for the insightful and helpful comments and suggestions, which are very important for further improving the manuscript. In the following, we will reply to the reviewers' comments point-by-point.

To Reviewer #1 and Reviewer#2:

We would like to thank the reviewers for the insightful comments on our manuscript. We deeply appreciate their constructive suggestions which is helpful for the improving of the manuscript. In the following, we will reply to these comments point-by-point.

Reviewers: page 3, line 48: "... concept that originated from ..." should be "... concept that originates from ...". Line 60: "...prototypes in studying the GP effect." Should be "...prototypes for studying the GP effect.". Line 74: "... GP effect in hydrogen exchange reaction..." should be "... GP effect in the hydrogen exchange reaction...".

Reply: We agree with the reviewers. We have revised the text accordingly. Thanks!

Reviewers: page 4, line 96: "hinders the understanding of detailed quantum dynamics." should be "hindering the understanding of the detailed quantum dynamics.". Line 97: "required in experimental probing the GP effect" should be "required for experimentally probing the GP effect". Line 98: "at a collision energy of 1.72 eV" should be "at the collision energy of 1.72 eV". Line 101: "DCSs ... was obtained" should be "DCSs ... were obtained".

Reply: Thanks! We agree with the reviewers. We have revised the text accordingly.

Reviewers: Page 5, line 121: "the data analysis method shown in ..." should be "the data analysis method is shown in ...". Line 123-124: "using the adiabatic quantum dynamics method with GP included." I think that here the PES which has been used should be mentioned and referenced (BKMP2 PES).

Reply: We agree with the reviewer. We have revised the text accordingly. And we have mentioned and referenced the BKMP2 PES in the relevant sections of the revised manuscript (as ref.40). Thanks!

Reviewers: Page 11, line 231: "approaches HD molecular" should be "approach the HD molecule". Line 232: "HD molecular stretches" should be "HD molecule stretches". Line 248: "we have..." should be "We have ...". Line 249: "... between directed abstraction ..." should be "... between direct abstraction ...".

Reply: Thanks for the comment. We have revised the corresponding text as suggested.

Reviewers: Page 12, line 264: "of an Nd: YAG laser" should be "of a Nd:YAG laser". Line 277: the BKMP2 PES is mentioned, but there is not a reference for it.

Reply: Thanks for the comment. We have revised the text. We have mentioned and referenced the BKMP2 PES in the relevant sections of the revised manuscript (as ref.40). Thanks!

Reviewers: Page 13, line 326: BaÑares should be Bañares.

Reply: We have revised the typo. Thanks very much!

To Reviewer #3:

We would like to thank this reviewer for the insightful and helpful comments on the manuscript. In the following, we will reply to the comments point-by-point.

Reviewer: After decades of effort, the geometric phase effect on elementary chemical reactions has only recently been documented, largely through work of some of these authors. However, such effects have been seen only at high energy in the region around the conical intersection (CI). In this beautifully-written manuscript, the present authors combine extraordinary experimental data obtained far below the CI, where two interfering paths have vastly different weights, with quantum scattering calculations to demonstrate clear evidence of the GP effect. The effect is seen in a narrow backscattered region for particular product states, and the contributing partial waves identified. It was a great pleasure to read this paper, and I certainly believe it merits publication in Nature Comm. I recommend publication as-is, although I have one small comment: I don't know what the reference to "And/∧" means (l. 138) and I don't see a discussion of this in the SI. Perhaps they could clarify this.

Reply: Thanks very much for the positive comment. We agree with the reviewer that "And/∧" should be clarified. To make it clearer, we have changed "And/∧" to “∧” (means “and”), which is used to represent two ro-vibrational states with similar energies.